

# Testing convolutional neural network based deep learning systems: a statistical metamorphic approach

Faqeer ur Rehman[1] and Clemente Izurieta[1,2,3]

[1] Gianforte School of Computing, Montana State University, Bozeman, Montana, United States
[2] Pacific Northwest National Laboratory, Richland, Washington, United States
[3] Idaho National Laboratory, Idaho Falls, Idaho, United States

## ABSTRACT

Machine learning technology spans many areas and today plays a significant role in addressing a wide range of problems in critical domains, *i.e.*, healthcare, autonomous driving, finance, manufacturing, cybersecurity, *etc*. Metamorphic testing (MT) is considered a simple but very powerful approach in testing such computationally complex systems for which either an oracle is not available or is available but difficult to apply. Conventional metamorphic testing techniques have certain limitations in verifying deep learning-based models (*i.e.*, convolutional neural networks (CNNs)) that have a stochastic nature (because of randomly initializing the network weights) in their training. In this article, we attempt to address this problem by using a statistical metamorphic testing (SMT) technique that does not require software testers to worry about fixing the random seeds (to get deterministic results) to verify the metamorphic relations (MRs). We propose seven MRs combined with different statistical methods to statistically verify whether the program under test adheres to the relation(s) specified in the MR(s). We further use mutation testing techniques to show the usefulness of the proposed approach in the healthcare space and test two CNN-based deep learning models (used for pneumonia detection among patients). The empirical results show that our proposed approach uncovers 85.71% of the implementation faults in the classifiers under test (CUT). Furthermore, we also propose an MRs minimization algorithm for the CUT, thus saving computational costs and organizational testing resources.

## INTRODUCTION

According to the World Health Organization (WHO), in 2019 alone, pneumonia (a pulmonary infection illness) was responsible worldwide for almost 740,180 (14%) of all deaths in children under the age of 5 (*World Health Organization, 2022*). It has affected children and families all around the globe, but the worst among them are mostly from sub-Saharan Africa and South Asia. One of the most promising diagnostic tools witnessed in medical science (for pneumonia detection) includes taking advantage of chest radiography but it remains a time-consuming and resource-intensive task for radiologists to manually examine and diagnose chest X-rays (*Lavine, 2012*). At the same time, manual approaches

Corresponding author
Faqeer ur Rehman,
faqeerrehman@msu.montana.edu

are error-prone and subjective. Therefore, this raises a strong need for efficient Artificial Intelligence (AI) based methods that can automatically detect the existence of pneumonia among patients correctly.

AI is playing a pivotal role in empowering and building solutions in critical domains, such as medical science, in which mostly deep learning (DL) based models, *i.e.*, convolutional neural networks (CNN), have become the state-of-the-art technique. Recent work has shown their usefulness/effectiveness in the detection of pneumonia using chest X-ray images (*Rajpurkar et al., 2017*; *Jaiswal et al., 2019*; *Nath & Choudhury, 2020*; *Tilve et al., 2020*). It is important to highlight that in the healthcare space (and in general), the research community is focusing more on the development of high-performance DL-based models (which is critical) but much less on performing their quality assurance (which is also a very important but unfortunately much-ignored area).

It is important to note that testing ML models presents additional challenges for software testing teams when compared to traditional software testing. First, when testing traditional software, the focus is usually on testing the source code. However, with ML-based models, testing the data adds an extra layer of complexity. Second, the decision logic is not explicitly hard-coded in ML-based models, but rather it is learned from the data to make inferences. Third, the input space for which the ML models must be verified is enormous, with features such as price, background color of an image, temperature, *etc.*, having a wide range of valid values. Therefore, testing the ML model for all possible feature values/scenarios is either impossible or extremely time-consuming (*Marijan, Gotlieb & Ahuja, 2019*). Last but not least, testing computationally complex ML models suffers from the Oracle problem. An *oracle or test oracle* is a mechanism that a software tester uses to verify whether the output produced by the program(s) under test (PUT) is correct or not. The Oracle problem arises when either the oracle is not available or is available but infeasible to apply within a limited set of constraints (*Barr et al., 2014*).

In the face of the above-highlighted challenges, it is essential to have effective testing strategies for testing ML-based models. It can be argued that ML engineers already use different performance evaluation metrics (*i.e.*, accuracy, precision, recall, F1-score, *etc.*) to perform some testing activities during the development of ML models. However, these evaluation metrics are not meant to test ML-based models to find implementation bugs in them. Instead, they are used to evaluate/validate which ML algorithm is best suited for the underlying data/problem. Furthermore, models with low-performance metrics scores often represent ML problems related to the availability or feeding of insufficient data. However, if the problem is not related to the insufficiency of data, and instead, there is an implementation fault in the DL-based classifier, then we need to understand the causes for the algorithm's low performance. Collecting more images, labeling them (a resource-intensive task), and feeding them to the same buggy algorithm will be of no advantage to further improving the model's performance unless we focus on new verification techniques. Ultimately, the ML engineer will start looking for alternative algorithms, wrongly assuming that the current algorithm used is not appropriate for addressing the underlying problem. Therefore, it is essential that we first make sure that the model under

test does not have any implementation bugs before collecting, labeling, and feeding more data into it.

Among the traditional software testing techniques, metamorphic testing (MT) (*Chen, Cheung & Yiu, 2020*) is considered a simple but very powerful approach in alleviating the Oracle problem when testing ML-based applications. Instead of checking the correctness of individual output (when the oracle in the form of expected output is not available), it generates a follow-up test case and then reasons about the relation between the source and follow-up inputs and the corresponding outputs generated for them. This relation is known as the metamorphic relation (MR), and the deviation from the relation is treated as an identification of a bug in the program under test.

When it comes to the verification (finding implementation bugs) of the CNN-based DL models using the MT approach, it brings an added challenge, *i.e.*, the stochastic nature of their training (because of randomly initializing the network weights) makes the application of traditional MT techniques infeasible. To better understand, suppose we have a CNN model $M$, training data $D$, and the test data $T$. If we train this CNN model $M$ multiple times on the same training data $D$ with the same network configuration and use those trained models to predict the output for the same test data $T$, we see slightly different outputs. This is due to the fact that each time a CNN model is trained, it gets initialized with some random weights that may make it converge at slightly different points, thus causing the model to predict different outputs for the same test instances in $T$. Therefore, if we apply the conventional MT technique in verifying the CNNs, the different outputs generated for the source and follow-up test cases during subsequent runs cannot be attributed to the identification of bugs in the model. Instead, the violation of MR is due to the stochastic nature of training in CNNs that generated different outputs for the subsequent runs. This raises an important question: 'How can we use MT to verify CNN-based classifiers in a non-deterministic environment?' The purpose is to verify that those models are implemented correctly and do not have any implementation bugs. One possible approach is to fix the random seeds so that each time the CNN model gets initialized with the same network weights, it predicts the consistent outputs for the given source and follow-up test instances. However, it is important to highlight that when the data is processed by DL system in parallel processes or if the underlying hardware architecture uses GPUs, it may still induce non-determinism in calculations when leveraged by libraries like Nvidia CUDA libraries for training CNNs; thus, fixing the random seeds will not necessarily ensure deterministic outputs across multiple runs (*Dwarakanath et al., 2018*). Second, even if the random seeds are fixed to enable deterministic mode, not only some of the popular APIs provided by Pytorch *i.e.*, 'MaxPool3d', 'AvgPool3d', 'AdaptiveMaxPool2d', 'EmbeddingBag' and 'CTCLoss' lack deterministic implementation and will generate run-time errors (*The Linux Foundation, 2025*) but also the training process for some popular networks *i.e.*, 'Seq2Seq' will not work if the PyTorch is set in a deterministic mode (*Qian et al., 2021*). Third, setting the deterministic mode may also adversely increase the model training time (*Qian et al., 2021*); thus, stretching the overall testing phase and ultimately delaying the model deployment to a production environment. Last but not least, in a real

environment, we also do not prefer fixing the random seeds for achieving better performance in CNNs. Therefore, it raises a strong need for the existence of effective testing strategies that cannot only verify the CNNs in their intended environment (in case an organization wants to leverage popular APIs with GPU capabilities for accelerating the training time) but also do not require the software tester to modify the ideal settings (configured by the software engineers) for the model submitted for testing. These limitations are the motivations for our work to propose an approach that can help verify CNN models in a stochastic environment.

It is noteworthy to mention that most of the existing literature focuses on using MT for validating deep learning models (*Ding, Kang & Hu, 2017*; *Wang & Su, 2020*; *Gudaparthi et al., 2021*; *Naidu, Gudaparthi & Niu, 2021*; *Yu et al., 2022*; *Bouchoucha et al., 2023*); however, there does exist few research studies available that use the MT technique to target the verification aspect of DL models, but they require a software tester to fix the random seed during the training phase; thus, they are not applicable to verify DL models in non-deterministic environment (*Dwarakanath et al., 2018*; *Li et al., 2020*). In this study, we address this problem by combining MT with hypothesis testing (called statistical MT) that uses multiple statistical measures to target the verification aspect of CNN-based pneumonia detection models in a stochastic environment. It is important to mention that our work is different from *Guderlei & Mayer (2007)*, who uses MT to construct new hypothesis tests and conducted a case study to test inverse cumulative distribution function in a simulation software. In our proposed approach, we obtain the results for multiple trained models instead of comparing the results for two subsequent runs (*i.e.*, one for the source data and one for the follow-up data). We then compare the results of the source test data with the follow-up test data using different statistical methods and verify whether the MRs are satisfied or not. A violation of MR will suggest the existence of a potential bug in the program under test. It is important to mention that to the best of our knowledge, we have not found any research work that focuses on using statistical MT techniques for 'verification' (finding the implementation bugs) of CNN-based image classifiers in a non-deterministic environment, especially in the healthcare domain.

In this article, we make the following contributions:

- A statistical MT technique is used to verify CNN-based image classifiers in a stochastic environment where the underlying root cause of model failure is an implementation bug. This is the first such research work that does not require a software tester to fix the random seeds during training to verify the output of DL models. To the best of our knowledge, the work of *Dwarakanath et al. (2018)*, *Li et al. (2020)* is the only work available in the literature that uses MT to target the verification aspect of testing deep learning models but their approach requires fixing the random seeds during training to test DL-based applications, which limits its applicability in real-world scenarios.
- We propose seven MRs that both the researchers and practitioners can leverage (in combination with the proposed statistical methods) to uncover implementation bugs in CNN-based image classifiers in a stochastic environment.

- We show the applicability of the proposed approach in the healthcare space by testing two CNN-based deep learning models that are used to detect pneumonia among patients. All the data, code, and results are open-sourced (https://github.com/matifkhattak/SMT4DL).

- We use the mutation testing technique to show the effectiveness of the proposed MRs, *i.e.*, their ability to detect implementation faults in the PUT. The results obtained show that using the proposed approach, we are able to uncover 85.71% of the injected faults in the PUT.

- In the context of the statistical metamorphic testing approach, we take advantage of using multiple concepts, *i.e.*, mutation scores, the type of mutants killed by each MR, and the difference of sets (borrowed from set theory) to propose an MRs minimization algorithm; thus, saving computational costs and organizational testing resources.

We expect that a software testers can use our approach as follows. When the CNN-based model is submitted for testing, our approach can be used to verify the program in an automated way. When using our approach, the software testers will be able to focus only on the testing of the program and should not worry about the tasks (*i.e.*, fixing random seeds across the project, identifying how to get deterministic outputs across multiple runs to verify the MRs, *etc.*) that may deviate them from the actual task, which is testing, and can lead to wastage of time in a strict time-constraint environment. They can select any of the MRs, and our approach will help automatically generate test cases to statistically verify the intended characteristics of the program under test. If any of the statistical MRs is violated, it will depict the identification of potential bugs in the program.

## BACKGROUND

This section briefly discusses the core concepts used in this work, *i.e.*, convolutional neural network, metamorphic testing, metamorphic relations, and mutation testing techniques.

### Convolutional neural network

CNN is a type of feed-forward neural network that is widely used in solving image-related problems (*Nisha & Meeral, 2021*). Its architecture consists of an input layer, hidden layers, and an output layer. The hidden layers may contain a series of convolution, pooling, and fully-connected layers, as shown in Fig. 1. Each convolution layer contains several filters, which are used to extract specific patterns from the input data. For example, given the images of animals, filters in the first layer may look for the low-level texture features, *e.g.*, edges, colors, *etc.*, whereas the filters in the next layer may extract the high-level features, *i.e.*, nose, eyes, *etc.* The filter is like a window that slides over the input image and calculates the inner product of weight parameters and the pixels of the input image. The activation function is then applied to the calculated dot product, and this information is forwarded as input to the next layer. It is important to note that as the training progresses, the weight parameters and the image filters will be adjusted during the backpropagation process (using the gradient-descent optimization algorithm). The pooling layer performs dimensionality reduction by sliding the window over the output received from the

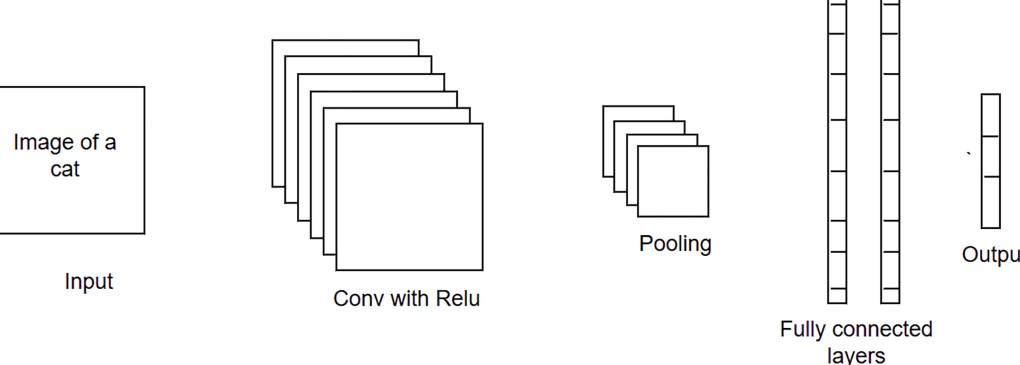

**Figure 1 Convolutional neural network (CNN) architecture.**

convolutional layer and obtaining the pooled value (either a maximum or an average value) for that sliding window. Lastly, the multidimensional data is flattened using the fully connected layer before feeding it to the output layer of the network. The output layer will use either the softmax (if the problem is a multi-class classification problem) or the sigmoid activation function (if the problem is a binary-class classification problem). The output layer produces the probability vector. The class for which the probability score is highest is considered as a final/predicted output for the given test instance.

## Metamorphic testing

In contrast to ML, software engineering (SE) is a more mature field; however, there is limited synthesized knowledge of SE's best methods and practices to develop better, test, operate, and maintain ML applications. Our focus in this research work is primarily on the testing aspects of ML-based models utilizing traditional SE testing techniques. We aim to make a contribution by exploring i) whether we can use any traditional software testing techniques directly to test ML models and ii) whether we can adapt techniques, and apply them to ML testing. Among the traditional software testing techniques, MT is a simple but very powerful approach to alleviating the oracle problem when testing ML-based applications. MT is a property-based testing strategy used to address the oracle problem in testing both supervised (*Dwarakanath et al., 2018*; *Li et al., 2020*; *ur Rehman & Izurieta, 2021b*; *Santos et al., 2020*; *ur Rehman & Izurieta, 2021a*; *Xie et al., 2011*) and unsupervised ML applications (*Rehman & Izurieta, 2022b, 2022a*; *Yang, Towey & Zhou, 2019*; *Xie et al., 2020*). MT captures a program's necessary characteristics/properties in the form of relations known as MRs that the PUT is expected to satisfy. Instead of checking the output for individual input, MT uses multiple executions to verify whether multiple inputs and their related outputs satisfy the relation specified in the MR. If the program does not adhere to the relation captured in the MR, it would suggest that some potential bug exists in the PUT. As shown in Fig. 2, a *source input* and a *follow-up input* (generated by performing a transformation on a source input) are used as inputs to the same PUT. If the

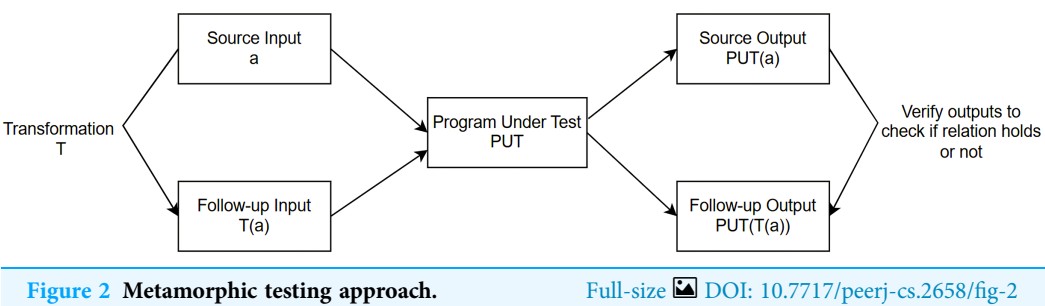

**Figure 2 Metamorphic testing approach.**

output for the *source input* and *follow-up input* does not satisfy the relation captured in the MR, the MR is said to be violated, and a potential bug is detected in the PUT.

## Mutation testing

Mutation testing is a fault-based testing technique (*DeMillo, Lipton & Sayward, 1978*) that is not only used to check the effectiveness of test cases and associated data but can also be used in simulating real faults made by software developers (*Andrews, Briand & Labiche, 2005*). An artificial bug (*i.e.*, a mutant) is injected into the source code of the original program **P** to generate a faulty program version **P′** in order to check whether the available test cases can kill the mutant or not. A mutant is killed if, for the same given test input, the output of the mutated program **P′** does not resemble the output generated by the original program **P**. The mutation score (the number of mutants killed/total number of mutants) is calculated to determine/identify the effectiveness of a test case. If a mutant cannot be killed with the existing test data, then the data set needs to be extended.

## RELATED WORK

The traditional testing activities performed while developing ML models are cross-validation and evaluation of the performance metrics (*i.e.*, accuracy, precision, recall, F1-score, *etc.*). However, they do not aim to test ML-based models for finding bugs in them. Instead, they are used to evaluate which ML algorithm is best suited for the underlying data/problem. Imagine there is an implementation fault in the underlying ML algorithm used for prediction. In that case, the aforementioned evaluation metrics will not provide any useful information about the existence of such a fault. *Xie et al. (2011)*, showed through empirical evaluation that the cross-validation alone is not enough in uncovering the implementation faults in the classifiers (*i.e.*, k-NN, and naive Bayes) under test. Furthermore, they show that MT is an effective testing technique in uncovering the buggy behavior of the mutated program versions that were generating a similar cross-validation pattern as that of the original/mutant free program version. *Li et al. (2020)* and *ur Rehman & Izurieta (2021a)* utilized mutation testing techniques to generate mutated program versions (for fully connected NN-based classifiers) that had almost the same high accuracy as that produced by the original program, knowing *a priori* that they represented the buggy versions of the PUT; thus, showing that i) using high accuracy as a testing criterion is not a reliable choice, and ii) a program producing high accuracy is not an indication that the PUT is free from bugs. Furthermore, the work in *Li et al. (2020)* proposes a set of MRs to

kill high accuracy producing mutants, whereas the work in *ur Rehman & Izurieta (2021a)* uses a statistical hypothesis testing technique for identification of high accuracy producing mutants and an ML-based approach for prediction of buggy behavior in the next release of a classifier(s) under test (CUT).

The MT approach has been widely applied across a number of domains, *i.e.*, data analytics, image captioning, self-driving vehicles, and natural language processing. *Jarman, Zhou & Chen (2017)* proposed an MT-based approach for testing Adobe Analytics software in which they tested an Adobe cloud-based time series analysis service (used for anomaly detection in marketing data) from both the verification and validation perspective. *Yu et al. (2022)* proposed MetaIC, an approach that uses MT for testing image-captioning (IC) systems. Their approach extracts an object and inserts it into a randomly selected image to form an image pair. The caption returned by the system under test (for the provided image pair) is verified to check if the MR is violated. The authors tested multiple systems (*i.e.*, Azure cognitive service and five other IC models) and identified numerous captioning errors in them, showing the effectiveness of MT in testing state-of-the-art IC systems. Similarly, *Wang & Su (2020)* proposed an MT-based approach for testing object detection models. Their approach first generates synthetic images by adding objects to an image treated as a background and then uses those images to verify the proposed MRs. The results obtained show that the proposed approach exposed erroneous behaviors for more than 38,000 input images in several commercial object detectors. *Zhang et al. (2018)* proposed DeepRoad, a GAN-based MT approach for testing self-driving systems. The authors argued that the existing research work (*Pei et al., 2017*; *Tian et al., 2018*) for testing autonomous driving systems has the drawback of generating unrealistic driving scenes that can never exist in real life, thus compromising their efficacy and reliability. The authors argue that the DeepRoad adds realistic weather conditions (*i.e.*, sunny, snowy, and rainy) in an image and effectively uncovers many erroneous behaviors in three DNNs under test. Similarly, *Guo, Feng & Chen (2022)* proposed an MT-based approach for testing LiDAR-based self-driving vehicles and identified several inconsistent results in four DL models. *Jiang et al. (2021)*, applied MT in the natural language processing (NLP) space. The authors proposed a set of six MRs for testing five natural language inference models. Their proposed approach not only detects many prediction errors but also helps the users interpret different characteristics of the models under test. *Yan, Yecies & Zhou (2019)* conducted a case study in the field of NLP where they proposed MRs for validating the quality of text messages. *Zhou, Xiang & Chen (2015)* leverage the MT technique for addressing the oracle problem in testing search engines. They proposed a set of 5 MRs and conducted a study to perform the quality assurance of popular search engines, *i.e.*, Google, Bing, Chinese Bing, and Baidu, from a user perspective (validation) that can help them decide whether or not the system meets their expectations, especially when they do not have access to the documentation and internal implementation details of the system. *Yuan et al. (2021)* tested the visual questioning answering system using the MT technique, in which they performed a set of transformations on the input images and the questions to test if the answers proposed by the system adhere to the proposed MRs. The

authors applied the proposed approach for testing 10 visual question-answering systems and exposed millions of inconsistencies in them.

The MT technique has also shown its usefulness in testing both supervised and unsupervised ML algorithms/applications. For testing unsupervised ML algorithms, *Yang, Towey & Zhou (2019)* proposed seven MRs to assess the behavior of the k-means clustering algorithm, provided by the WEKA tool (https://ml.cms.waikato.ac.nz/weka/), whereas *Xie et al. (2020)* proposed 11 MRs targeting users' general expectations from six unsupervised ML algorithms. However, both the research studies used simple synthetic 2D data and only targeted the 'validation' aspects of testing the clustering algorithms. These limitations were recently addressed in *Rehman & Izurieta (2022b, 2022a)*, where the authors proposed a diverse set of MRs targeting both the 'verification' (finding implementation bugs) and 'validation' (meeting the general user's expectations) aspects of testing clustering algorithms. The authors used multi-dimensional real-world data sets to test density-based, hierarchy-based, and prototype-based unsupervised ML algorithms. Last but not least, we refer interested readers to consult recent surveys (*Chen et al., 2018*; *Segura et al., 2016*) focused on metamorphic testing, its advantages and applications, its ability to address the oracle problem in a diverse range of applications, and the open challenges for the research community to address.

In contrast to testing unsupervised ML applications, the MT technique has received more attention when testing supervised ML applications. *Santos et al. (2020)* utilized MT to validate a k-NN based breast cancer classifier and showed the applicability of MRs (originally proposed by *Xie et al. (2011)*) to test k-NN and naive Bayes algorithms in the healthcare domain. *Naidu, Gudaparthi & Niu (2021)* conducted a short survey in which they first extracted a set of MRs proposed in the space of autonomous driving, then performed their categorization into five groups, and finally used them for 'validating' three different CNN-based models being trained on ImageNet dataset (*Deng et al., 2009*). Similarly, the work of *Gudaparthi et al. (2021)* primarily focused on i) discussing various test case prioritization techniques for the MRs, and ii) testing the reliability of CNN-based image classification models using the MT approach. *Ding, Kang & Hu (2017)* apply the MT technique for testing a deep learning framework. They proposed a set of 11 MRs that primarily focus on testing the 'validation' aspect of the framework by expecting that the deep learning framework should produce better accuracy than the SVM. *Dwarakanath et al. (2018)* proposed four MRs for verifying an SVM-based model (used for classification of hand-written digit images), whereas four MRs were proposed to verify a DNN-based image classifier called a residual network (ResNet), used for addressing a multi-class classification problem. The results show that, on average, their approach can uncover 71% of implementation faults. As mentioned earlier, one of the challenges of applying MT to CNN-based architecture, *i.e.*, ResNet, is their non-deterministic nature in training and producing different outputs for the same inputs across multiple runs. So, to obtain deterministic results across subsequent runs, the authors fixed the random seeds for verification of the proposed MRs. The authors acknowledged that fixing the random seeds fixed the non-determinism on CPUs but not on GPUs. Therefore, they conducted all their

experiments on CPUs. It is important to mention that their work resembles ours only in the sense that their approach also focuses on the 'verification' (*i.e.*, finding implementation bugs) aspect of CNNs. However, their approach uses the conventional MT technique to test CNNs in an environment, *i.e.*, CPUs, where the random seed could be fixed to get deterministic outputs over subsequent runs for verification of the MRs, whereas our approach can be applied to verify the CNNs in their non-deterministic environment using the proposed set of statistical MRs. *Li et al. (2020)* handled a similar stochastic behavior in testing the NN-based classifier by initializing the random weights with constant values, so that consistent results could be obtained (for both the source/original and the generated follow-up inputs) to verify the proposed MRs. However, similar to *Dwarakanath et al. (2018)*, their research work also leaves few questions unaddressed, *i.e.*, what if, i) it is not possible/feasible for a software tester to manually identify and fix the random seeds for a large number of libraries integrated with the project, ii) a third party library does not provide any options to fix random seeds, and, iii) the underlying hardware architecture uses GPUs that may induce non-determinism in calculations when leveraged by libraries, *i.e.*, Nvidia CUDA Libraries (*Dwarakanath et al., 2018*). In summary, the existing literature has two main focuses: i) it emphasizes the use of conventional MT to target the validation aspect of testing deep learning-based image classifiers, and ii) while the work of *Dwarakanath et al. (2018)* and *Li et al. (2020)* use MT for the verification of DL-based models, their approach of fixing the random seed is not applicable to test them in a stochastic environment, which hinders the applicability of their approaches in real-world scenarios. To the best of our knowledge, we haven't found any work that primarily focuses on showing the usefulness of using statistical MT to uncover implementation bugs in CNN-based classifiers in a non-deterministic environment, which we have proposed in this work.

Approaches like statistical hypothesis testing has been used to test software with random outputs but either they rely on knowing beforehand the theoretical characteristics of output or reference implementation (*Guderlei et al., 2007*; *Ševčíková et al., 2006*), which are not always available. *Guderlei & Mayer (2007)* work focuses on addressing these limitations by using MT to construct new hypothesis tests and conducted a case study to test inverse cumulative distribution function in a randomized software, which has further been applied in testing stochastic optimisation algorithms (*Yoo, 2010*), and simulation software (*Ahlgren et al., 2021*). Most recently, *Underwood, Luu & Liu (2023)* proposed MT framework that uses hypothesis testing for validating non-deterministic automated driving systems. However, their work focuses on the 'validation' aspect of automated driving systems, not the 'verification' aspect, which aims to uncover implementation bugs in the implementation under test. In this study, our focus is to address the aforementioned challenges associated with the 'verification' of non-deterministic CNN-based image classifiers, which is an area that the existing literature seriously lacks. We use statistical MT technique in combination with mutation testing to test such computationally complex programs without worrying about either fixing the random seed or initializing the weights (of neural network) with constant/fixed values.

# APPROACH TO IDENTIFY IMPLEMENTATION BUGS IN CNN-BASED DL APPLICATIONS

In this section, we provide an in-depth analysis of our proposed statistical metamorphic testing technique for testing two CNN-based DL models used for the identification of pneumonia among patients using chest X-ray images. We propose a set of seven MRs that have been used in combination with statistical methods for uncovering implementation bugs in a CNN-based image CUT. As mentioned earlier, due to the stochastic nature of training in CNNs, two subsequent runs cannot be used to verify the relation specified in the MR. Therefore, in our proposed approach, we obtain the results for multiple runs of the model (for both the source and follow-up test data) and then statistically compare them to verify whether or not the relation specified in the MR is satisfied. Empirical evidence from *Dwarakanath et al. (2018)* shows that for different deep learning architectures, a correct NN classifier might have different converge points for multiple runs, but those will be very close to each other and will not differ significantly in terms of overall loss.

If the model under test is already a trained model, then in the context of MT, the source and follow-up data will point to the test instances. However, in our case, we target the 'verification' aspect of the classifiers under test, in which we need to train/test the model multiple times in order to verify the MRs using the proposed statistical methods. In that case, source data is the original data comprising the training and test data, and the follow-up data (generated from source data) is also comprised of the training and test data. That is why for a better understanding, we treat source/follow-up data as a combination of training source/follow-up data (used to train the models), and the source/follow-up test data (used to test the trained models).

To further elaborate the proposed approach, as shown in Fig. 3, we use the MRs to generate the follow-up training/validation/test data from the source training/validation/test data. Then, using the source training data and the follow-up training data, we train the CUT multiple times and use all trained models to predict the outputs for the same source and follow-up test data. The results obtained are statistically analyzed (with the help of proposed statistical methods) to verify the satisfiability of the relation captured in the MR. An MR is said to be violated if there is a significant difference in the outputs generated by the models for the given source and follow-up test instances, which would ultimately be a sign of a potential bug in the CUT. To show the effectiveness of the proposed MRs, we further use the mutation testing technique for injecting artificial bugs in the CNN-based CUT, and then empirically evaluate the number of bugs uncovered by the proposed approach. Lastly, an MRs minimization algorithm is executed to identify a minimum set of MRs that would have the same fault detection effectiveness as using the full set of seven MRs.

## Statistical metamorphic relations

We propose a set of seven MRs, some of them are modified from *Dwarakanath et al. (2018)*, *Naidu, Gudaparthi & Niu (2021)* for the verification of two CNN-based pneumonia detection models under test that use chest X-ray images for the detection of pneumonia among patients. We call these MRs as 'Statistical MRs' as they are verified through

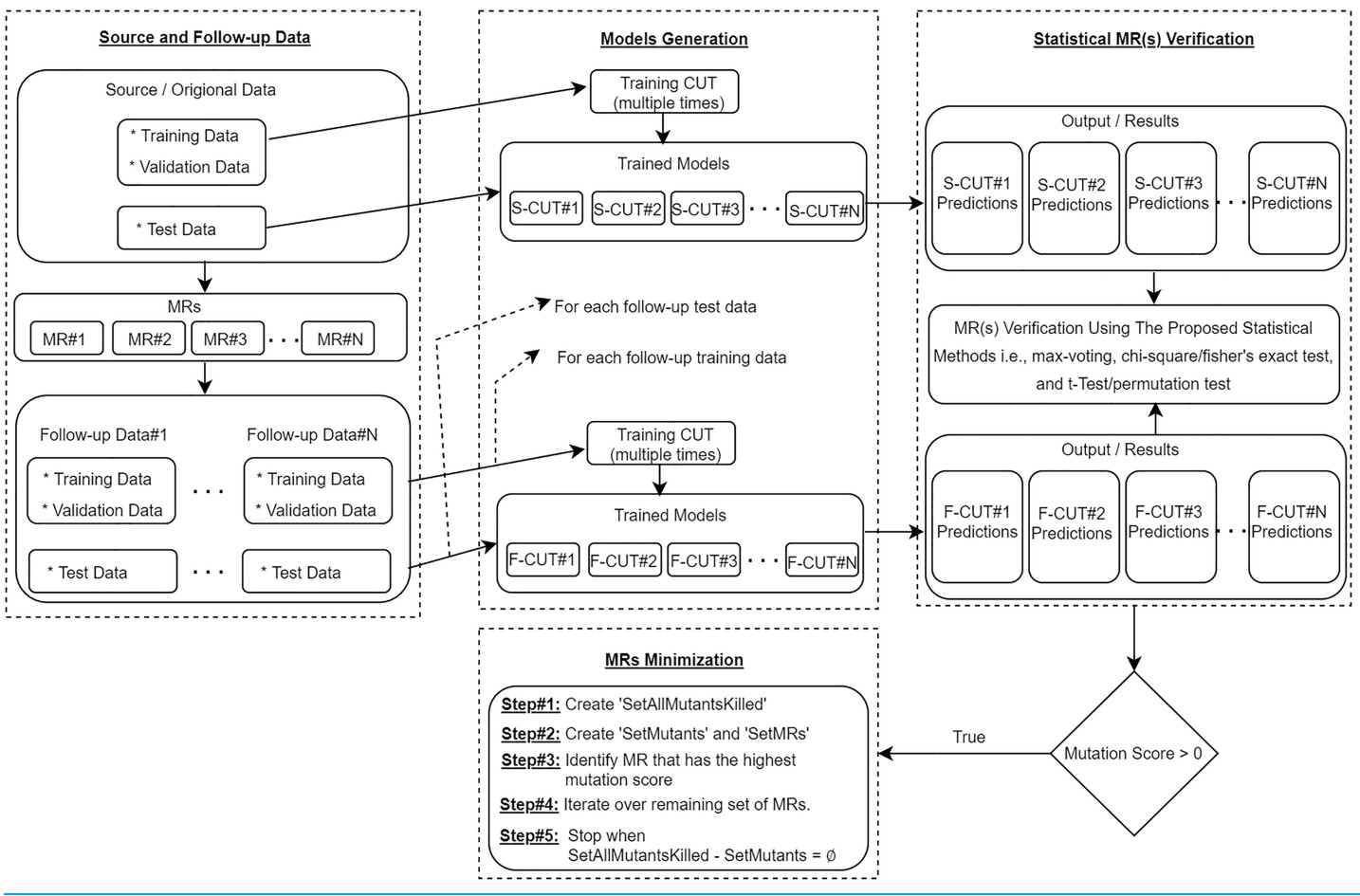

**Figure 3** Proposed statistical metamorphic testing approach.               

statistical approaches. These MRs target the necessary characteristics expected from the classifiers under test. Before conducting any experiments, we performed a verification step to check that the transformations proposed in the MRs are valid and do not significantly affect the overall performance (*i.e.*, accuracy and F1-score) of the original/mutant free model.

### MR-1: Blurring the X-ray images

Suppose, we have the training data represented as $X_{train}$, validation data as $X_{validation}$, and the test data as $X_{test}$. We train the CUT multiple times on the source training data $X_{train}$ and use those trained models for prediction on the source test data $X_{test}$. This MR specifies that if we slightly blur the X-ray images to generate follow-up training, validation, and test data, the difference in the results obtained for the given follow-up test instance $x_{test}^i$ (in the follow-up test data) over multiple runs should not be statistically significant from the results obtained for the same test instance $x_{test}^i$ in the source test data. This MR uses the Gaussian blur (with a radius of the blur effect = 1, representing the standard deviation of the Gaussian kernel) filtering technique to smooth the images, and the model is expected to

be robust for such small changes and should not significantly change the results for the input data.

### MR-2: Flipping the X-ray images

Suppose we have the training data represented as $X_{train}$, validation data as $X_{validation}$, and the test data as $X_{test}$. We train the CUT multiple times on the source training data $X_{train}$ and use those trained models for prediction on the source test data $X_{test}$. This MR specifies that if we flip the X-ray images vertically (through the y-axis) to generate the follow-up training, validation, and test data, it will not change the existing relationship between the image pixels. Therefore, the difference in the results produced by the program for the given test instance $x_{test}^i$ (in the follow-up test data) over multiple runs should not be statistically significant from the results obtained for the same test instance $x_{test}^i$ in the source test data.

### MR-3: Mirroring the X-ray images

Suppose we have the training data represented as $X_{train}$, validation data as $X_{validation}$, and the test data as $X_{test}$. We train the CUT multiple times on the source training data $X_{train}$ and use those trained models for prediction on the source test data $X_{test}$. This MR specifies that if we mirror the X-ray images (through the x-axis) to generate the follow-up training, validation, and test data, the difference in the results produced by the program for the given test instance $x_{test}^i$ (in the follow-up test data) over multiple runs should not be statistically significant from the results obtained for the same test instance $x_{test}^i$ in the source test data. Similar to flipping an image, this MR also preserves the relationship between the pixels of an object, and a correct CNN-based model should learn from the relationship among pixels, not from their position in space.

### MR-4: Adding a small rectangle (outside the region of interest) to the X-ray images

Suppose we have the training data represented as $X_{train}$, validation data as $X_{validation}$, and the test data as $X_{test}$. We train the CUT multiple times on the source training data $X_{train}$ and use those trained models for prediction on the source test data $X_{test}$. This MR specifies that if we add a small rectangle near the border of an X-ray image, *i.e.*, (outside the region of interest, *i.e.*, chest) to generate the follow-up training, validation, and test data, the difference in the results obtained for the given test instance $x_{test}^i$ (in the follow-up test data) over multiple runs should not be statistically significant from the results obtained for the same test instance $x_{test}^i$ in the source test data. In theory, a correct image classification model should depend on the target (*i.e.*, chest), not on a small rectangle, which is the same across all images, and it is also not the differentiating factor for the models to learn.

### MR-5: Rotating the X-ray images

Suppose we have the training data represented as $X_{train}$, validation data as $X_{validation}$, and the test data as $X_{test}$. We train the CUT multiple times on the source training data $X_{train}$ and use those trained models for prediction on the source test data $X_{test}$. This MR specifies that if we rotate the X-ray images (*i.e.*, by 180 degree) to generate the follow-up training, validation, and test data, the difference in the results produced by the program for the

given test instance $x_{test}^i$ (in the follow-up test data) over multiple runs should not be statistically significant from the results obtained for the same test instance $x_{test}^i$ in the source test data.

### MR-6: Adding scattered dots to the X-ray images

This MR can be used to simulate the scenario when an X-ray machine has either some dust on it or is malfunctioning. Suppose we have the training data represented as $X_{train}$, validation data as $X_{validation}$, and the test data as $X_{test}$. We train the CUT multiple times on the source training data $X_{train}$ and use those trained models for prediction on the source test data $X_{test}$. This MR specifies that if we randomly scatter some dots (in our case, between 300–1,000) of a single pixel size across the image to generate the follow-up training, validation, and test data, the difference in the results obtained for the given test instance $x_{test}^i$ (in the follow-up test data) over multiple runs should not be statistically significant from the results obtained for the same test instance $x_{test}^i$ in the source test data.

### MR-7: Sharpening the X-ray images

This MR can simulate a situation when the relevant department slightly sharpens an X-ray image (without degrading the image quality) to enhance its quality and better identify disease among patients. Suppose we have the training data represented as $X_{train}$, validation data as $X_{validation}$, and the test data as $X_{test}$. We train the CUT multiple times on the source training data $X_{train}$ and use those trained models for prediction on the source test data $X_{test}$. This MR specifies that if we slightly sharpen the X-ray images (making them more focused) to generate the follow-up training, validation, and test data, the difference in the results produced by the program for the given test instance $x_{test}^i$ (in the follow-up test data) over multiple runs should not be statistically significant from the results obtained for the same test instance $x_{test}^i$ in the source test data.

## Statistical MRs verification methods

This section discusses the proposed statistical MT approach we used to verify/check the relation(s) captured in the proposed MR(s). As previously mentioned, the limitation of using the conventional MT technique is that it does not work well when there is non-determinism in the outputs of a program (such as CNNs, because of randomly initializing the network weights). Therefore, we use a statistical MT technique that statistically compares the outputs (using the statistical measures presented in *ur Rehman & Izurieta (2021b)*) over multiple iterations and verify whether the difference in the results is statistically significant or not. If it is significant, the null hypothesis stating that *'For a given test instance $x_{test}^i$, the difference in the outputs is not statistically significant'* is rejected. This ultimately suggests that the relation captured in the corresponding MR is violated and indicates the existence of a potential bug in the application under test.

### Maximum voting

In order to apply the maximum voting concept, we train the model 'n' times on the given source training data $X_{sd}$ and 'n' times on the generated follow-up training data $X_{fd}$. This results in producing $M_n$ number of trained models on $X_{sd}$ and $M_n$ number of trained

**Table 1 Frequency distribution table for class labels 'Normal' vs 'Pneumonia'.**

| Execution type | Normal | Pneumonia |
|---|---|---|
| Source ($n = 30$) | 5 | 25 |
| Follow-up ($n = 30$) | 7 | 23 |

models on $X_{fd}$. We then compare the results to see whether, for the given test instance $x_{test}^i$, the class which is predicted maximum number of times by the models trained on source training data is different from the class predicted maximum number of times by the models trained on the follow-up training data. If so, the MR is said to be violated.

### Comparing distributions over all class labels

One of the limitations we observed for the maximum voting approach is that there may exist some cases where the class predicted the maximum number of times by the models trained on the source and follow-up training data is the same, but there is a significant difference in the frequency distribution over all predicted class labels. In such scenarios, the maximum voting concept may not provide realistic results in identifying bugs. Therefore, this provides further motivation to compare the frequency distributions over all class labels (instead of just one) for better identification of faults in the PUT. We formulate this problem as comparing all the class labels' frequency distributions obtained for the models trained on the source training data (treated as the expected distribution) with the class labels' frequency distribution obtained for the models trained on the follow-up training data (treated as the observed distribution) and create 2 * 2 contingency table, as shown in Table 1. This type of data can now be statistically compared using the Chi-square parametric test of homogeneity ($\tilde{\chi}^2$) (*Ramsey & Schafer, 2012*). However, one of the assumptions for the Chi-square test of homogeneity is that each cell value (in a frequency distribution table) should have a reading of at least five (*McDonald, 2014*). In the cases where this assumption was violated, we also applied Fisher's non-parametric exact test, which does not require this assumption to be satisfied. In order to compare the distributions using the Chi-square test of homogeneity/Fisher's exact test, we propose the following null and alternative hypothesis:

- $H_0$: There is no difference in the class distributions obtained for the models trained on the source and follow-up training data.
- $H_a$: The class distribution obtained for the models trained on the source training data is different from the class distribution obtained for the follow-up training data.

  If the $p$-value obtained is below the set significance level, the null hypothesis is said to be rejected, which will suggest that there is a significant difference in the obtained class distributions. Thus, the MR is said to be violated.

### Comparing distributions over probability scores

Instead of comparing the final output/class labels for the test instances, another method we propose is to statistically verify the MRs by comparing the predicted probability scores for

| Table 2 Exemplary probabilities. | |
| --- | --- |
| **Source executions** | **Follow-up executions** |
| 0.77 | 0.57 |
| 0.71 | 0.51 |
| 0.79 | 0.53 |
| 0.72 | 0.52 |
| 0.75 | 0.53 |
| 0.72 | 0.53 |

the given test instances. This method will be beneficial in identifying scenarios when the probability scores over predicted classes (for the source and follow-up test data) have been significantly changed by the models. Unlike the Chi-square/Fisher exact test, this method will allow us to compare the outputs at a more granular level to identify the faults in the CUT better. In the CNN-based CUT, the last layer (which is an output layer) uses the sigmoid activation function, which produces a probability vector containing probability scores for all the class labels. For the given test instance $x^i_{test}$, the class for which the probability score is higher is treated as the final output of the model. We leverage these probability scores and perform statistical analysis to check whether, for the given test instance $x^i_{test}$, the probability scores predicted by the models trained on the source data are close to the scores predicted by the models trained on the follow-up data. If the difference is found statistically significant, the MR is said to be violated.

In order to compare the probability scores, we developed a Windows-based desktop utility (using C#) that processes the raw results (obtained in the form of probability scores for each class label) and transforms them into a form on which statistical analysis could be performed. We first identify the class label that is predicted the maximum number of times (*i.e.*, identifying the class for which the model is most likely to be sure) by the models trained on the source training data. We then leverage this knowledge and extract the probability scores for the same class predicted by the models trained on the follow-up training data. As an example, Table 2 shows the extracted probability scores for a specific class predicted by both types of models, *i.e.*, models trained on the source training data and the models trained on the follow-up training data.

We treat this problem as comparing two sample means and propose the following null and alternative hypotheses:

- $H_0$: There is no difference in the sample means of probability scores.
- $H_a$: There is a significant difference in the sample means of probability scores.

This problem can now be treated as comparing two sample means for which the parametric t-test (*Ramsey & Schafer, 2012*) is used to verify if the two groups are significantly different from each other. However, one of the t-test assumptions is the normality assumption's satisfiability. Although the t-test is considered robust to slight violations in the normality assumption but during our analysis, we found some cases

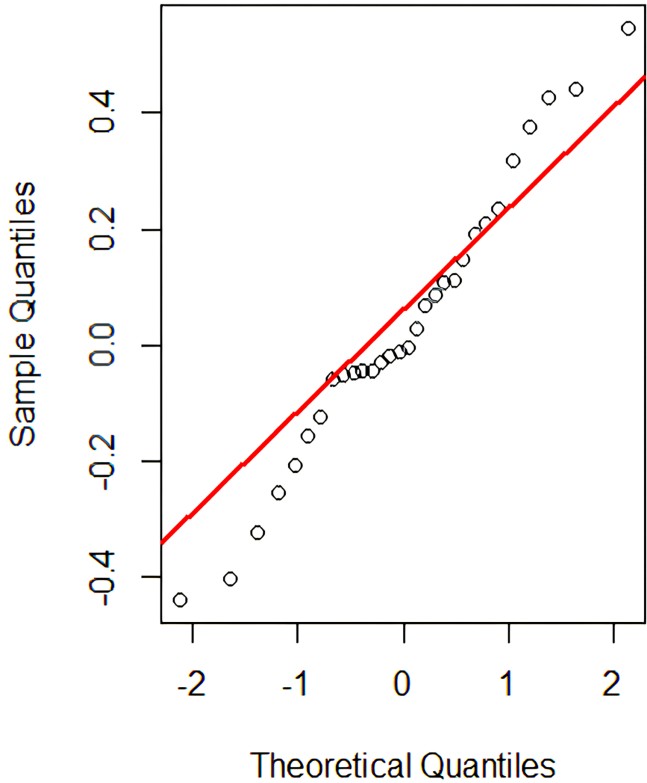

**Figure 4  Q-Q plot.**

where this assumption is badly violated. For example, Fig. 4 shows the Q-Q plot (for MR-5) that has long tails at the end, showing a slight violation of the normality assumption. Therefore, for better analysis and decision-making purposes, we also applied the non-parametric 'permutation test' (to present our results), which does not rely on the normality assumption to be satisfied (*Ramsey & Schafer, 2012*). For the given test instance $x^i_{test}$, if the difference between the probability scores is found to be statistically significant (*i.e.*, less than the set significance level), the null hypothesis $H_0$ is rejected; thus, the MR is said to be violated, which signifies the existence of a potential bug in the PUT.

## EXPERIMENTAL SETUP

All the experiments were conducted on a machine running Windows 11, having 16 GB RAM, and an AMD Ryzen 7 5825U processor with eight cores. We evaluate our proposed statistical metamorphic testing technique for testing two CNN-based pneumonia detection models. Both the CNN-based classifiers under test are implemented using python 3.9 on top of Keras 2.8.0 and TensorFlow 2.8.0 libraries. As shown in Table 3, the CNN architecture used in the first open source CUT (https://github.com/sanghvirajit/Medical-Image-Classification-using-CNN) is comprised of five convolutional blocks. In total, the network consists of 27 layers, among which 10 are convolutional, four are dense layers, and 13 layers are of type batch normalization, dropout, and max-pooling. The batch normalization and dropout can be considered as regularization techniques, where the former helps in stabilizing the training process and reducing the internal covariate shift,

**Table 3 CNN architectures for applications under test.**

|  | Total layers | No of conv layers | No of dense layers | No of other layers | Activation functions | Batch size | Epochs | Optimizer |
|---|---|---|---|---|---|---|---|---|
| App#1 | 27 | 10 | 4 | 13 | Relu, Sigmoid | 32 | 10 | Adam |
| App#2 | 12 | 5 | 2 | 5 | Relu, Sigmoid | 32 | 10 | Adam |

and the latter helps in preventing overfitting problems by randomly deactivating a certain number of neurons. The network uses an 'adam' optimizer with 'binary_crossentropy' loss to adjust network weights to minimize the difference between actual and predicted outputs. The output of the last convolutional block is flattened into a 1-dimensional vector, and then a series of fully connected layers with dropout layers are used. All the convolutional and fully connected layers use the Relu activation function. The output layer uses the sigmoid activation, which generates the probability scores (between 0 and 1) for the given classes and predicts the class for which the probability score is higher. Similarly, the second CNN architecture we implemented comprises five convolutional layers, each separated by the max-pooling layer that helps reduce the input dimensions and retain the most important features. In the end, a flattening layer is used, followed by a couple of fully connected layers. All the fully connected and convolutional layers use the Relu activation function, whereas the output layer uses the sigmoid activation function. The network's hyperparameters include an 'adam' optimizer with 'binary_crossentropy' loss, uses ten epochs for training, and a batch size of 32.

The validation method used for both models under test is the hold-out validation method, in which the portion of validation data is held out, and the rest of the data is used to train the model. At the end of each epoch, the model's performance is evaluated on a validation set to see how well the model is generalizing to unseen data. Both models use two callback functions to monitor performance and adjust the training process based on certain criteria. The first callback function is used to monitor the validation loss, and if it does not improve for two consecutive epochs, the learning rate is reduced by a factor of 0.3. Similarly, the second one also monitors the validation loss but is used to stop training early if the validation does not improve by at least 0.1 for one epoch.

As shown in Table 4, our training dataset is comprised of 5,216 chest X-ray images, 440 validation instances, and 624 test instances, labeled with 'normal' and 'pneumonia', which have been taken from https://github.com/sanghvirajit/Medical-Image-Classification-using-CNN/tree/main/chest_xray. In order to inject mutants into the code responsible for training and testing the classifiers, we have leveraged popular mutation testing tool called 'MutPy' (https://pypi.org/project/MutPy/). However, one of the limitations of such tools is that they can't inject mutants inside the libraries code *i.e.*, Keras, and Tensorflow, so a manual effort was required to inject mutants at random locations. All the injected mutants in the applications under test were of type 'first order mutants'. In total, 103 mutants were injected, among which most of them were either invalid or crashing the models under test. Apart from that, few of them (injected by MutPy tool) were of type equivalent mutants (*Ammann & Offutt, 2016*), which was very hard and time consuming task to find them

**Table 4 Dataset description.**

| Class labels | Train | Validation | Test | Domain |
| --- | --- | --- | --- | --- |
| Normal | 1,341 | 142 | 234 | Healthcare |
| Pneumonia | 3,875 | 298 | 390 | |
| **Total** | **5,216** | **440** | **624** | |

```
apply_state = self._prepare(var_list)

    if experimental_aggregate_gradients:

        grads_and_vars = self._transform_unaggregated_gradients(grads_and_vars)

        grads_and_vars = self._aggregate_gradients(grads_and_vars)

        grads_and_vars = self._transform_gradients(grads_and_vars)
```

Original Code

```
apply_state = self._prepare(var_list)

    if experimental_aggregate_gradients:

        grads_and_vars = self._transform_unaggregated_gradients(grads_and_vars)

        # grads_and_vars = self._aggregate_gradients(grads_and_vars)

        grads_and_vars = self._transform_gradients(grads_and_vars)
```

Mutated Code

**Figure 5 An example of the 'Statement Removal' type mutant.**

manually. After excluding all those mutants, we ended up using 14 valid mutants. For example, one of the mutants is shown in Fig. 5, which prevents model from aggregating the calculated gradients and may cause multiple models to converge at significantly different points. To check complete list of mutants, we refer the interested readers to visit[1].

Due to the stochastic nature of training CNN-based CUTs, we use multiple iterations to obtain the results for the source, and follow-up test cases to statistically verify the MRs. Empirical evidence from *Dwarakanath et al. (2018)* shows that for different deep learning architectures, a correct NN classifier might have different converge points for multiple runs, but those will be very close to each other and will not differ significantly in terms of overall loss. For each MR, we train the model 'n' times on the given source/original data

[1] The entire set of valid mutants are here: https://github.com/matifkhattak/SMT4DL.

**Table 5 Results for maximum voting Concept, ✓ denotes the mutant is killed.**

**App#1**

| | MR1 | MR2 | MR3 | MR4 | MR5 | MR6 | MR7 |
|---|---|---|---|---|---|---|---|
| Mutant1 | ✓ | | | | | ✓ | |
| Mutant2 | | | | ✓ | | | |
| Mutant3 | | | | | | | |
| Mutant4 | | | | ✓ | | | |
| Mutant5 | | ✓ | ✓ | ✓ | | ✓ | ✓ |
| Mutant6 | | | | ✓ | | ✓ | |
| Mutant7 | ✓ | | | | | | |
| **MutationScore** | 28.57% | 14.29% | 14.29% | **57.14%** | 0% | 42.86% | 14.29% |
| | | | | **85.71%** | | | |

**App#2**

| | MR1 | MR2 | MR3 | MR4 | MR5 | MR6 | MR7 |
|---|---|---|---|---|---|---|---|
| Mutant1 | ✓ | ✓ | | ✓ | ✓ | ✓ | |
| Mutant2 | ✓ | ✓ | | | ✓ | | |
| Mutant3 | | | | | | | |
| Mutant4 | ✓ | ✓ | ✓ | | ✓ | ✓ | |
| Mutant5 | ✓ | ✓ | ✓ | ✓ | ✓ | ✓ | |
| Mutant6 | | | | | | ✓ | |
| Mutant7 | | ✓ | | | ✓ | ✓ | |
| **MutationScore** | 57.14% | **71.43%** | 28.57% | 28.57% | **71.43%** | **71.43%** | 0% |
| | | | | **85.71%** | | | |

**Note:**
Bold text represents the MR(s) that killed highest percentage of mutants.

and 'n' times on the generated follow-up data (where, $n = 30$); thus, the total sample size = 60. We then use these trained models to predict the output for the given source and follow-up test data. All the necessary information about the predicted outputs (*i.e.*, test instance, predicted class label, expected class label, probability vector, and max probability for the predicted class) is logged into Microsoft Excel files. We then develop a .Net Framework-based desktop utility (using C#) to process this raw data further and transform/store it into a form on which we apply the proposed statistical methods (implemented in R) to verify the relation specified in the MRs. The violation of the MR will suggest that the mutant is killed. The CNN models, their code and architecture, details about the injected mutants, the MRs implementation, the raw Excel files and their processed versions, the .Net utility and R scripts to process raw output files, and the final results are all publicly available in a shared GitHub repository (https://github.com/matifkhattak/SMT4DL).

We use the mutation score, which is the 'Number of mutants killed/Total number of injected mutants' as a measure to show the effectiveness of our proposed approach in the identification of faults injected in the CUT. As shown in Tables 5–8, we use this measure to not only show the results of each MR at an individual level but also for combined MRs for each of the proposed statistical methods. We use a standard significance level ($\alpha$) of 0.05 and statistically evaluate the results to check whether the difference is statistically

**Table 6 Results for chi-square test/fisher exact test (significance level ($\alpha$) = 0.05), ✓ denotes the mutant is killed.**

**App#1**

|  | MR1 | MR2 | MR3 | MR4 | MR5 | MR6 | MR7 |
|---|---|---|---|---|---|---|---|
| Mutant1 |  | ✓ | ✓ | ✓ | ✓ | ✓ |  |
| Mutant2 |  |  | ✓ | ✓ |  |  |  |
| Mutant3 |  |  |  |  |  |  |  |
| Mutant4 |  | ✓ | ✓ | ✓ |  |  |  |
| Mutant5 | ✓ |  |  |  |  | ✓ |  |
| Mutant6 |  |  |  | ✓ |  | ✓ |  |
| Mutant7 |  |  | ✓ |  |  |  |  |
| **MutationScore** | 14.29% | 28.57% | **57.14%** | **57.14%** **85.71%** | 14.29% | 42.86% | 0% |

**App#2**

|  | MR1 | MR2 | MR3 | MR4 | MR5 | MR6 | MR7 |
|---|---|---|---|---|---|---|---|
| Mutant1 |  | ✓ | ✓ | ✓ | ✓ |  |  |
| Mutant2 | ✓ | ✓ |  | ✓ | ✓ |  |  |
| Mutant3 |  |  |  |  |  |  |  |
| Mutant4 |  |  |  | ✓ | ✓ |  |  |
| Mutant5 |  |  |  | ✓ |  |  |  |
| Mutant6 |  |  |  |  |  | ✓ |  |
| Mutant7 |  | ✓ |  | ✓ | ✓ |  |  |
| **MutationScore** | 14.29% | 42.86% | 14.29% | **71.43%** **85.71%** | 57.14% | 14.29% | 0% |

**Note:**
Bold text represents the MR(s) that killed highest percentage of mutants.

significant or not. If it is, the *Null* hypothesis is said to be rejected, suggesting that the MR has been violated and the mutant is killed.

## Goal question metric approach

The goal question metric (GQM) is a well-known technique (*Basili, 1992*) used for the identification of goal-oriented measures. We use it to frame our research work. As shown in Fig. 6, we first identify a set of goal(s), followed by the identification of research question(s) used to refine the goal(s) further, and then define/identify the metrics used as measures to answer the research question(s) in a quantifiable way.

**Research goal (RG):** To detect buggy behavior in supervised ML models for the purpose of improving their quality from the perspective of software engineer in the context of testing CNN-based models in a non-deterministic environment.

**Research Question 1 (RQ1):** How effective is the statistical metamorphic testing approach in the verification (identification of implementation bugs) in the CUT?

**Research Question 2 (RQ2):** Do all MRs (verified through different statistical methods) show the same fault detection ability?

**Research Question 3 (RQ3):** Using the proposed approach, which MR(s) have high fault detection effectiveness, and which among them has the least?

**Table 7 Results for t-test (significance level ($\alpha$) = 0.05), ✓ denotes the mutant is killed.**

**App#1**

|  | MR1 | MR2 | MR3 | MR4 | MR5 | MR6 | MR7 |
|---|---|---|---|---|---|---|---|
| Mutant1 |  | ✓ | ✓ | ✓ | ✓ | ✓ | ✓ |
| Mutant2 | ✓ | ✓ | ✓ | ✓ | ✓ | ✓ | ✓ |
| Mutant3 |  |  |  |  |  |  |  |
| Mutant4 | ✓ | ✓ | ✓ | ✓ | ✓ | ✓ | ✓ |
| Mutant5 | ✓ |  |  | ✓ |  | ✓ |  |
| Mutant6 | ✓ | ✓ | ✓ | ✓ | ✓ | ✓ | ✓ |
| Mutant7 | ✓ | ✓ | ✓ | ✓ | ✓ | ✓ | ✓ |
| **MutationScore** | 71.43% | 71.43% | 71.43% | **85.71%** | 71.43% | **85.71%** | 71.43% |
|  |  |  |  | **85.71%** |  |  |  |

**App#2**

|  | MR1 | MR2 | MR3 | MR4 | MR5 | MR6 | MR7 |
|---|---|---|---|---|---|---|---|
| Mutant1 |  | ✓ | ✓ | ✓ |  |  |  |
| Mutant2 | ✓ | ✓ | ✓ | ✓ | ✓ |  | ✓ |
| Mutant3 |  |  |  |  |  |  |  |
| Mutant4 |  | ✓ | ✓ | ✓ | ✓ |  | ✓ |
| Mutant5 |  |  | ✓ | ✓ |  | ✓ |  |
| Mutant6 | ✓ | ✓ | ✓ | ✓ | ✓ | ✓ | ✓ |
| Mutant7 | ✓ | ✓ | ✓ | ✓ | ✓ | ✓ | ✓ |
| **MutationScore** | 42.86% | 71.43% | **85.71%** | **85.71%** | 57.14% | 42.86% | 57.14% |
|  |  |  |  | **85.71%** |  |  |  |

**Note:**
Bold text represents the MR(s) that killed highest percentage of mutants.

**Table 8 Results for permutation test (significance level ($\alpha$) = 0.05), ✓ denotes the mutant is killed.**

**App#1**

|  | MR1 | MR2 | MR3 | MR4 | MR5 | MR6 | MR7 |
|---|---|---|---|---|---|---|---|
| Mutant1 |  | ✓ | ✓ | ✓ | ✓ | ✓ | ✓ |
| Mutant2 | ✓ | ✓ | ✓ | ✓ | ✓ | ✓ | ✓ |
| Mutant3 |  |  |  |  |  |  |  |
| Mutant4 | ✓ | ✓ | ✓ | ✓ | ✓ | ✓ | ✓ |
| Mutant5 | ✓ |  |  | ✓ |  | ✓ |  |
| Mutant6 | ✓ | ✓ | ✓ | ✓ | ✓ | ✓ | ✓ |
| Mutant7 | ✓ | ✓ | ✓ | ✓ | ✓ | ✓ | ✓ |
| **MutationScore** | 71.43% | 71.43% | 71.43% | **85.71%** | 71.43% | **85.71%** | 71.43% |
|  |  |  |  | **85.71%** |  |  |  |

**App#2**

|  | MR1 | MR2 | MR3 | MR4 | MR5 | MR6 | MR7 |
|---|---|---|---|---|---|---|---|
| Mutant1 |  | ✓ | ✓ | ✓ |  |  |  |
| Mutant2 | ✓ | ✓ | ✓ | ✓ | ✓ |  | ✓ |
| Mutant3 |  |  |  |  |  |  |  |
| Mutant4 |  | ✓ | ✓ | ✓ | ✓ |  | ✓ |

| Table 8 (continued) | | | | | | | |
|---|---|---|---|---|---|---|---|
| **App#2** | | | | | | | |
| Mutant5 | | | ✓ | ✓ | | ✓ | |
| Mutant6 | ✓ | ✓ | ✓ | ✓ | ✓ | ✓ | ✓ |
| Mutant7 | ✓ | ✓ | ✓ | ✓ | ✓ | ✓ | ✓ |
| **MutationScore** | 42.86% | 71.43% | **85.71%** | **85.71%** | 57.14% | 42.86% | 57.14% |
| | | | | **85.71%** | | | |

**Note:**
Bold text represents the MR(s) that killed highest percentage of mutants.

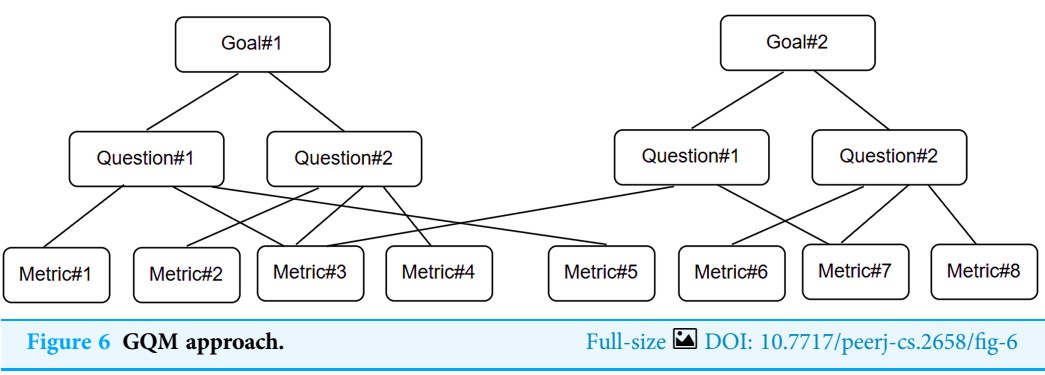

**Figure 6  GQM approach.**

**Research Question 4 (RQ4):** How can the proposed statistical MRs be minimized for the CNN-based pneumonia detection models under test?

**Research Metric 1 (RM1):** *Mutation score (Overall)*—Percentage of mutants killed using the proposed approach.

**Research Metric 2 (RM2):** *Mutation score (per MR)*—Percentage of mutants killed by each MR.

## EMPERICAL RESULTS AND DISCUSSION

**RQ1: How effective is the proposed statistical metamorphic testing approach in the verification (identification of implementation bugs) in the CUT?**

In Table 5, we provide the mutation score (used as a metric to answer the research questions in a quantifiable manner) for all the combined MRs verified through the maximum voting concept, Table 6 provides the mutation score for all the combined MRs verified through the chi-square test/fisher exact test. Table 7 provides the mutation score for all the combined MRs verified through the t-test, and Table 8 provides the mutation score for all the combined MRs verified through the permutation test. We inject seven valid mutants into each of the CUT, among which six (85.71%) have been killed by the proposed MRs (for all the statistical methods). So, the overall fault detection effectiveness of the proposed MRs for both the applications under test is 85.71%, suggesting that the proposed statistical metamorphic testing approach effectively identifies faults in the CUT, which answers RQ1.

**RQ2: Do all MRs (verified through different statistical methods) have the same fault detection effectiveness?**

In Tables 5–8, we also present the mutation score of the MRs at the individual level. The results in Table 5, and 6 show that for both the applications under test, most of the MRs have different fault detection effectiveness when verified through the maximum voting concept and chi-square/fisher exact test. However, when the MRs are verified through T-Test/Permutation test (as shown in Tables 7, and 8), five out of seven MRs have the same fault detection effectiveness for App#1, whereas, for App#2, most of the MRs detects a different percentage of mutants, which answers RQ2 that in most cases different MRs have different fault detection effectiveness. This also suggests that the effectiveness of MRs varies from application to application, and it is better to use a diverse set of MRs for testing, especially if each has a different ability to uncover different types of faults in the PUT. Apart from that, the results obtained show that overall, individual MRs verified through the t-test/permutation achieve better results. This is because of the proposed statistical MRs verification approach that compares outputs at a more granular level *i.e.*, by comparing probability scores over predicted classes. Although this method is more effective in uncovering the injected bugs, the addition of an extra layer, *i.e.*, identifying the class for which the model is more confident and then extracting the probability scores for that specific class for both the source and follow-up instances from the raw output data, has its own extra computation cost.

**RQ3: Using the proposed approach, which MR(s) have high fault detection effectiveness, and which among them have the least?**

We analyze the results at a broader level to answer this research question. We collectively analyze the results provided in Tables 5–8 in order to understand how many of the total mutants have been killed by each of the individual MRs, assuming that the organization has enough testing resources to verify the MRs using all the proposed statistical methods. In this case, a mutant is said to be killed if any of the MRs (verified through any of the proposed statistical methods) has been violated. The results from Tables 5–8 are accumulated and presented in Table 9. We observe that for App#1, all the MRs except MR5 achieve high fault detection effectiveness, and noticed a couple of interesting cases: i) MR5 is unable to kill mutant5 using any of the proposed statistical methods, and ii) none of the proposed MRs is able to kill mutant3. Similarly, for App#2, we conclude that i) none of the MRs kill mutant3, ii) MR7 has the least fault detection effectiveness, and iii) except MR6 and MR7, the rest of the MRs achieve high fault detection effectiveness, which answers RQ3. We perform further analysis to check why mutant3 has survived. Upon close analysis, we identified that all the trained CNNs have predicted the same class label and produced the same probability vector for all the given test instances; thus, none of the MRs were able to kill the mutant3. An important takeaway here is to realize and admit that some specific type of mutant(s) may exist that can force the model to converge at exactly the same point, no matter how many times it is trained on the same data. In such types of scenarios, the model will generate consistent (but wrong) outputs for both the source and follow-up test inputs; thus, the MR will falsely be satisfied (false positive) and wrongly report that the model adheres to the relation specified in the

**Table 9  Cumulative results of MRs, ✓ denotes the mutant is killed.**

**App#1**

|  | MR1 | MR2 | MR3 | MR4 | MR5 | MR6 | MR7 |
|---|---|---|---|---|---|---|---|
| Mutant1 | ✓ | ✓ | ✓ | ✓ | ✓ | ✓ | ✓ |
| Mutant2 | ✓ | ✓ | ✓ | ✓ | ✓ | ✓ | ✓ |
| Mutant3 |  |  |  |  |  |  |  |
| Mutant4 | ✓ | ✓ | ✓ | ✓ | ✓ | ✓ | ✓ |
| Mutant5 | ✓ | ✓ | ✓ | ✓ |  | ✓ | ✓ |
| Mutant6 | ✓ | ✓ | ✓ | ✓ | ✓ | ✓ | ✓ |
| Mutant7 | ✓ | ✓ | ✓ | ✓ | ✓ | ✓ | ✓ |
| **MutationScore** | 85.71% | 85.71% | 85.71% | 85.71% | 71.43% | 85.71% | 85.71% |
|  |  |  |  | **85.71%** |  |  |  |

**App#2**

|  | MR1 | MR2 | MR3 | MR4 | MR5 | MR6 | MR7 |
|---|---|---|---|---|---|---|---|
| Mutant1 | ✓ | ✓ | ✓ | ✓ | ✓ | ✓ |  |
| Mutant2 | ✓ | ✓ | ✓ | ✓ | ✓ |  | ✓ |
| Mutant3 |  |  |  |  |  |  |  |
| Mutant4 | ✓ | ✓ | ✓ | ✓ | ✓ | ✓ | ✓ |
| Mutant5 | ✓ | ✓ | ✓ | ✓ | ✓ | ✓ |  |
| Mutant6 | ✓ | ✓ | ✓ | ✓ | ✓ | ✓ | ✓ |
| Mutant7 | ✓ | ✓ | ✓ | ✓ | ✓ | ✓ | ✓ |
| **MutationScore** | 85.71% | 85.71% | 85.71% | 85.71% | 85.71% | 71.43% | 57.14% |
|  |  |  |  | **85.71%** |  |  |  |

**Note:**
Bold text represents the MR(s) that killed highest percentage of mutants.

MR. It is worth mentioning to highlight that this is an inherent limitation of MT, *i.e.*, if the MR is satisfied, it does not necessarily mean that the PUT is free from bug(s), so with ours.

**Discussion:** The power of statistical test is the P (rejecting $H_o|H_o$ is false), which also means P (not making Type II error). The Type II error means 'accepting $H_o$ when it is false', and in the context of software testing, it will suggest that there is no bug in the program when in reality, there is. A high statistical power indicates a higher likelihood of detecting a true difference between groups if it exists. There are a couple of ways to increase the power of tests, which are under humans' control, *i.e.*, increasing the significance level $\alpha$ and the sample size 'n.' In Table 9, it can be seen that a few mutants, *i.e.*, mutant5 in App#1, mutant1 in App#2, mutant2 in App#2, and mutant5 in App#2 are not killed by some of the MRs that could be the symbol of an existence of Type II errors. In this case, one possible way to increase the power of test/decrease Type II error is to increase the $\alpha$. However, the drawback of increasing the $\alpha$ is that it will also increase the P (Type I) error. So, there is a trade-off in increasing/decreasing the $\alpha$. If a researcher thinks that Type II error is more severe and is willing to make this trade-off, he/she can increase the $\alpha$. On the contrary, if the severity of Type I error is higher, the $\alpha$ can be decreased accordingly. So, it depends on the nature of the application and the nature of the problem under investigation that how the knob of $\alpha$ is adjusted. Therefore, given the importance of detecting faults in

software testing, we have used a significance level of 0.05 to minimize potential false positives, which is appropriate and it is also a very common choice among researchers conducting statistical studies (*Ramsey & Schafer, 2012*). The other better approach to increase the power of tests is to increase the sample size, *i.e.*, the higher, the better. According to the central limit theorem, if we increase the sample size, the sampling distribution of the sample mean tends to become normally distributed. Usually, a sample size greater than 30 is considered adequate for performing reliable statistical analysis (*Ramsey & Schafer, 2012*). Due to time constraints and limited resources, we conducted 60 training runs (*i.e.*, sample size = 60) to verify each MR that we think is enough to capture the behaviors of trained models. However, one can further increase the sample size based on the availability of resources in the organization.

**Question#1:** Why are the Chi-square and Fisher's exact tests selected to compare the frequency distributions over predicted classes?

As shown in Table 1, for a given test instance, the predictions obtained for the categorial variables (class labels) are used to create 2 * 2 contingency table, which represents the frequency distribution over predicted classes. As we are interested in comparing all the class labels' frequency distributions obtained for the models trained on the source training data (treated as the expected distribution) with the class labels' frequency distribution obtained for the models trained on the follow-up training data (treated as the observed distribution), we need to identify the statistical test that is best fit for dealing with such type of data. Since we have the categorial variables and every data point in our result set is an independent observation, *Ramsey & Schafer (2012)* propose using either the Chi-square test or Fisher's exact test for analyzing such data (arranged in the 2 * 2 contingency table) to check if the difference between frequency distribution over classes/variables is statistically significant or not. It is important to mention that parametric tests are generally considered more reliable because they rely on certain assumptions that must be satisfied. Therefore, we selected using the Chi-square test of homogeneity, which is a parametric test and requires some assumptions to be satisfied. The first assumption, 'independence of observations' is satisfied in our case because the results/predictions obtained from each trained model are independent of other trained models. The second assumption 'the data should be categorial' is also satisfied because both the class labels (normal and pneumonia) are categorical variables. Similarly, the assumption of 'random sampling' is satisfied because we did not fix any random seed; instead, the results are obtained based on the random initialization of network weights. However, in some cases, the last assumption *i.e.*, 'each cell value (in a frequency distribution table) should have a reading of at least five' was not satisfied. So, for those cases, we applied Fisher's non-parametric exact test, which is a non-parametric test applicable to analyze such type data arranged in a 2 * 2 contingency table and does not require this assumption to be satisfied.

**Question#2:** Why are the t-test and permutation test chosen for comparing the probability scores of two groups?

As mentioned earlier, to perform statistical analysis for a given test instance at a more granular level, *i.e.*, using probability scores predicted for class labels, we extract this

information from the probability vectors produced by the trained models, as shown in Table 2. The data can be viewed/treated as two samples/groups having continuous values (because the sigmoid activation function used in the classification layer produces probability scores between 0 and 1). For a given test case, the first group comprises results from 30 models trained on the source data, whereas the other group comprises results from 30 models trained on the follow-up data. To statistically compare the sample means of two groups with continuous data between 0 to 1, and assuming the observations are independent, the recommended parametric test is the t-test (*Ramsey & Schafer, 2012*). While tests like ANOVA and MANOVA are suitable for comparing sample means for more than two groups, the t-test is the appropriate choice for our case. However, t-test has certain assumptions, *i.e.*, independent samples/observations, random samples, and normality assumption, which must be satisfied. In our experiment, assumptions like independent samples/observations and random were satisfied but for some of the test cases, the normality assumption was violated (as shown in Fig. 4). Although t-test is robust to such a violation, to perform a better and more reliable analysis, we applied 'permutation test', which is a non-parametric test and does not rely on normality assumption (*Ramsey & Schafer, 2012*). We would like to mention that we did not observe any major differences in the results obtained for t-test and permutation test. So, the choice of applying a non-parametric test solely depends on the organization, assuming they have enough resources available to use. We want to mention that performing statistical analysis of results at individual test instances' level offers a significant advantage. Consider we have a total of 10 test instances, out of which, for three of them, the MR is violated for the mutated version of CUT. This not only helps us identify which specific MR is violated but also pinpoints the exact test instances that reveal the fault. In the future, such fault-revealing instances can become part of prioritized test instances that the organization can use to perform regression testing. This level of insight would not be possible if we were to collectively compare the results of all test instances.

Someone may argue about deciding which statistical test (among all the proposed statistical measures) is better to use. When selecting a specific statistical test, the decision largely depends on the level of analysis someone is interested in. For example, suppose software testers are interested in analyzing the results at the class prediction level (*i.e.*, comparing frequency distributions over multiple predicted classes). In that case, tests like the chi-square test, fisher exact test, or max-voting will be the best choice. On the other hand, if the focus is to compare the results at a more granular level, *i.e.*, using class probabilities to statistically verify MRs, tests like t-test/permutation would be more appropriate. Additionally, if the organization has sufficient testing resources, it can perform analysis using both class predictions and probability scores for a more comprehensive comparison and improved results. Our experience shows that comparing results at a more granular level, *i.e.*, comparing class probabilities using the t-test, is more beneficial. Still, an extra cost is attached to processing the raw results and extracting the class probabilities.

---

**Algorithm 1** Statistical MRs' minimization algorithm.

**Input:** MRsDictionary<str,set>                    ▷/* consists of <MR, mutants it killed<

**Output:** Prioritized set of MRs

1: **while** all injected faults are covered **do**

2:     *selectedMRs* ← select MR(s) from *MRsDictionary* that killed maximum no of mutants

3:     **if** selectedMRs.count() = 1 **then**                    ▷/* just one MR has identified maximum number of mutants

4:         /* Add MR to the prioritized set.

5:     **end if**

6:     **if** selectedMRs.count() > 1 **then**                    ▷/* multiple MRs have identified same number of mutants

7:         **if** all MRs have killed same type of mutants **then**

8:             /* Randomly select one of them and add it to prioritized set of MRs

9:         **end if**

10:        **if** all MRs have killed different type of mutants **then**

11:            /* Add each of the MRs to a separate set of prioritized ordering

12:        **end if**

13:    **end if**

14: **end while**

15: **return** prioritized set containing minimum number of MRs

---

**RQ4: How can the proposed statistical MRs be minimized for the CNN-based pneumonia detection models under test?**

To answer this research question (in the context of the proposed statistical metamorphic testing approach), we take advantage of using a few concepts together, *i.e.*, the mutation score, the type of mutants killed by each MR, and the difference of sets (from set theory) to propose an MRs minimization algorithm. The proposed Algorithm 1 is used for finding the minimized set of MRs that are expected to have the same fault detection effectiveness (*i.e.*, the same mutation score) as that of using the original set of 7 MRs (*i.e.*, D), where $|C| <= |D|$; thus, saving computational costs and organizational testing resources. It is important to mention that our proposed MRs' prioritization algorithm is different from the work of *Srinivasan & Kanewala (2022)*, who propose fault based MRs prioritization algorithm in the context of traditional MT which does not uses any statistical measures to verify the relation captured in the MRs. Second, in their proposed approach, if they identify a list of MRs uncovering the same number of faults, they randomly choose one of them and add it to the prioritized MR ordering. However, an important question unaddressed is what if they all identify the same but different nature of faults? which may lead to identification of less effective prioritized set of MRs. In our proposed algorithm, we address this problem by incorporating the diversity of uncovered faults that helps in identification of more effective set of prioritized MRs.

**Discussion:** Consider a case for App#1 when the MRs are verified through the maximum voting concept, and we aim to find the minimum set of MRs for the

---

**Table 10 Output of statistical MRs minimization algorithm.**

**App#1**

| Priority | Max-voting | Chi-square/Fisher exact test | t-Test | Permutation test |
|---|---|---|---|---|
| 1 | MR4 | MR3—MR4 | MR4 or MR6 | MR4 or MR6 |
| 2 | MR1 | MR6—MR3 | | |
| 3 | | —MR6 | | |

**App#2**

| | | | | |
|---|---|---|---|---|
| 1 | MR5 | MR4 | MR3 or MR4 | MR3 or MR4 |
| 2 | MR6 | MR5 | | |

proposed approach. First, the algorithm identifies the MR that has the highest mutation score, *i.e.*, MR4, adds it to the *selectedMRs i.e.*, {MR4}, and the mutants it killed to *selectedMRMutants i.e.*, {Mutant2, Mutant4, Mutant5, Mutant6}. In case there are multiple MRs having the same mutation score, two possibilities can exist: i) they killed the same type of mutants. If so, randomly select any of them (since using one of them is enough to kill all those mutants), or ii) they killed different types of mutants. In this scenario, we incorporate the diversity aspect of MRs and treat each of them as a separate case to be handled. It is very important to incorporate this knowledge into the proposed MRs' prioritization algorithm, otherwise, a simple random selection of an MR can ultimately lead to less effective set of prioritized MRs. For example, in Table 6, it can be seen that for the Chi-square test/fisher exact test, MR3 and MR4 have the same mutation score but have killed different type of mutants. Therefore, two possible MRs prioritization paths could be initiated (*i.e.*, one starting with the MR3, whereas the other starting with MR4) that will end up in the identification of two MRs minimization sets, *i.e.*, {**MR3**, MR6} and {**MR4**, MR3, MR6} (as shown in Table 10). However, the set containing a minimal set of MRs is selected, *i.e.*, {MR3, MR6}, as a final set of prioritized MRs. Now consider, instead of initializing two MRs prioritization paths, what if we had just randomly selected MR4 between MR3 and MR4 (as proposed by *Srinivasan & Kanewala (2022)*, since they identified same number of faults), we would have been ended up with selecting less efficient set of prioritized MRs *i.e.*, {**MR4**, MR3, MR6}, which has been tackled by our proposed algorithm.

The results in Table 10 show that for App#1, when the MRs are verified through the maximum voting concept, just two MRs with their prioritization order MR4, and M1 are enough to attain a mutation score of 85.71% (*i.e.*, the overall mutation score achievable through all the seven MRs). Similarly, when verified through the chi-square/fisher exact test, the MR3, and MR6 are enough to attain the overall mutation score, whereas, for the MRs verified through the t-test and permutation test, only one MR (either of MR4 or MR6) is enough to kill the six mutants. For App#2, when the MRs are verified through the maximum voting concept, just two MRs with their prioritization order MR5, and M6 are enough to attain a similar mutation score achievable through all the seven MRs. Similarly, when the MRs are verified through the chi-square/fisher exact test, MR4, and MR5 are

enough to attain the overall mutation score, whereas, for the MRs verified through the t-test and permutation test, only one MR (either of MR3 or MR4) is enough to kill the 6 mutants, which answers RQ4. The minimized set of MRs will help in saving organizational resources and perform testing with fewer MRs (especially in a regression testing environment) without compromising the overall fault detection effectiveness of the proposed approach. A more comprehensive study to extend the proposed approach is in progress, which aims to leverage machine learning techniques for automatic identification/ prediction of a minimum set of MRs in a more efficient way.

## THREATS TO VALIDITY

### Internal validity

- We recognize that the proposed MRs' list is not exhaustive but they have been very useful in showing the effectiveness of the proposed statistical MT approach in uncovering a significant percentage of injected mutants in CNN-based classifiers under test. In the future, we aim to further expand the list to cover more complex scenarios.

### Construct validity

- One may argue that in real life, the minimized set of MRs may only be suitable for testing just the initial few releases/versions of the PUT and may not remain effective as the program evolves, which can be a potential threat to construct validity. However, we argue that the proposed MRs minimization algorithm can still be used for developing a more sophisticated data mining-based approach for addressing this challenge, *i.e.*, by preparing historical data over time using the output obtained through the proposed MRs minimization algorithm and then continuously mining that data for identifying an updated set of minimized MRs for better testing of the next release(s) of the PUT.

### External validity

- The effectiveness of proposed statistical MT approach is shown using mutation testing technique, a well known fault-based testing technique used for simulating real faults made by software developers (*Andrews, Briand & Labiche, 2005*). Although, the number of injected mutants may be low, the widely used mutation testing tool 'MutPy' has been employed to inject as many mutants as possible into the programs under test. Therefore, we believe that the proposed approach can be generalized to identify faults of similar nature in CNN-based classification models.
- Although we have shown the applicability of the proposed statistical MT approach in a medical image classification domain, the proposed approach is applicable to verify DL-based image classifiers in multiple other domains as well (*i.e.*, object detection in autonomous driving, product classification in retail business, defect detection in manufacturing, crops classification in agriculture, and property classification in real estate) in a stochastic environment, *i.e.*, on GPUs, using APIs that lack deterministic implementation, and use cases where we do not want to fix random seeds to get better model performance. There are a couple of reasons to strengthen

this claim. First, the proposed MRs are focused on the transformation of images and thus apply to the image classification domain in general. Second, the proposed statistical measures (used for verification of MRs) do not rely on any specific DL model architecture; hence, they can be used to verify any DL-based image classification application that has a stochastic nature in its results. Further empirical validation by extending the proposed approach to different domains helps mitigate threats to external validity.

## CONCLUSION

Manual examination of chest X-ray images (for detection of pneumonia among patients) is not only a time-consuming and resource-intensive task but is also subjective and prone to errors. This raises a need for an efficient method that can automatically detect the existence of pneumonia among patients correctly. ML-based solutions are playing a significant role in the automatic and timely detection of pneumonia among patients using chest X-ray images. To test such critical systems, the ML community frequently uses performance evaluation metrics, *i.e.*, accuracy, precision, recall, F1-score, *etc*. However, the fact is that these evaluation metrics are not meant to test (*i.e.*, verify and validate) ML-based models to find bugs in them; instead, they are used to evaluate which ML algorithm is best suited for the underlying data/problem. Apart from that, these evaluation metrics say nothing about the existence of faults in the model or the underlying algorithm. Therefore, using a systematic testing strategy is essential to test such critical systems to verify their correctness and enhance trust in them.

The metamorphic testing technique is a simple but effective testing strategy to alleviate the oracle problem in testing such computationally complex programs where the output for individual input is difficult to verify. However, the stochastic nature of the training of deep learning-based models (*i.e.*, the CNN-based pneumonia detection models in our case) adds an extra challenge and makes the conventional metamorphic testing technique infeasible to apply. In this research, we address this problem by using a statistical metamorphic testing technique that does not require fixing the random seeds to get deterministic results for verification of the MRs. We propose seven MRs in combination with statistical methods (for MRs verification) to verify two CNN-based image classifiers that have a stochastic nature in their results. The empirical results obtained show that our proposed method is able to uncover 85.71% of the implementation faults and can be used in testing non-deterministic ML classifiers. Furthermore, we also propose an algorithm for minimization of the proposed MRs that incorporates the diversity of mutants being killed, thus saving computational costs and organizational testing resources.

In the future, we aim to show further applicability of the proposed approach in testing different classes of CNN-based ML models used for addressing multi-class classification problems in the healthcare space. We also aim to further expand the MRs repository and leverage data mining/machine learning techniques for their effective prioritization and minimization in a more intelligent way.

## ACKNOWLEDGEMENTS

We thank the creator of the Medical Image Classification using CNN GitHub repo for making this dataset publicly available and for their efforts to support the research community.

### Funding

The authors received no funding for this work.

### Competing Interests

The authors declare that they have no competing interests.

### Author Contributions

- Faqeer ur Rehman conceived and designed the experiments, performed the experiments, analyzed the data, performed the computation work, prepared figures and/or tables, authored or reviewed drafts of the article, and approved the final draft.
- Clemente Izurieta conceived and designed the experiments, analyzed the data, authored or reviewed drafts of the article, and approved the final draft.

### Data Availability

The data is available at GitHub:

- https://github.com/sanghvirajit/Medical-Image-Classification-using-CNN/tree/main/chest_xray.

The code, data, raw results, data processor utility, and the final results we obtained are available at GitHub and Zenodo:

- https://github.com/matifkhattak/SMT4DL.

- Faqeer ur Rehman. (2025). matifkhattak/SMT4DL: v1.1.0 (v1.1.0). Zenodo. https://doi.org/10.5281/zenodo.14616977.

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
