# Peer review of "Testing convolutional neural network based deep learning systems: a statistical metamorphic approach"

_PeerJ Computer Science, doi:10.7717/peerj-cs.2658_

## Round 0.1 · original submission · Major Revisions

· Academic Editor

Major Revisions

Dear Authors,

Please carefully check the issues raised by the reviewers, especially Reviewer 2.
Moreover, the main aspects of novelty of your work as well as your contribution with respect the state of the art should be clarified since, in my opinion, it is not totally clear.

Best regards,

M.P.

Reviewer 1 ·

Basic reporting

In this paper, the authors reported their recent work on applying the so-called statistical metamorphic testing technique into the testing of deep learning based pneumonia detection systems. Seven MRs were defined, and mutation analysis was utilized to evaluate the performance of the technique. The experimental results demonstrated the applicability of the metamorphic testing technique in the specific field.

Experimental design

The experiments were mainly conducted in the field of healthcare. They were designed in a scientific way.

Validity of the findings

The experimental results support the claim made by the authors.

Reviewer 2 ·

Basic reporting

While the paper is generally well-written, some sections, especially around statistical methods and technical details, could benefit from clearer language to enhance accessibility for a broader audience.

Experimental design

The paper does not provide extensive details about the experimental setup, which may hinder replication of the study by other researchers. More specific information on the training and testing environments, model configurations, and hyperparameters would be beneficial.

Validity of the findings

The paper uses statistical methods to verify metamorphic relations, but the choice of statistical tests and the justification for their use could be more thoroughly explained. Additionally, the discussion of statistical power and potential Type II errors could be expanded.

Annotated reviews are not available for download in order to protect the identity of reviewers who chose to remain anonymous.

Reviewer 3 ·

Basic reporting

no comments.

Experimental design

no comments.

Validity of the findings

no comments.

Additional comments

This paper proposes a statistical metamorphic testing approach, for testing CNN based models that exhibit non-deterministic characteristics. The motivation behind this work is explained and justified. The approach and the supporting experiments are clearly clarified.

Strength: Good writing
Novel statistical metamorphic testing approach
Clear discussion of related work
Weakness: Insufficient interpretation of the experimental results

(1) The introduction of MT and MRs are not accurate enough. I suggest that you improve Section “Metamorphic Testing” on page 4.
Line 189-190, the description about the MR and source/follow-up test cases need to be improved. It is not right to state “each MR is comprised of a source and a follow-up test cases”
Intrinsically, MR specify the relationships among relevant inputs and their outputs.
Also, Figure 2 needs to be improved in order to avoid misunderstanding about MT and MRs.

(2) I think the combining MT with statistical test is one of the key novelties of this work.
Although each test is described, I still suggest you to further reason about them. Why different statistical tests are used for different comparisons?
For example, at line 460-461, “This type of data can now be statistically compared using the Chi-square parametric test of homogeneity”. Why Chi-square test can be applied here?
Secondly, it is unclear whether each test is performed on the results for each test instance (i.e., Xi-test declared at line 483) or is performend on the results of all test instances?
If it is the former case, how to make a decision based on different restuls from different instances (i.e., some are significant different, some are not). If it is the latter case, it needs to be explictely clarified.

(3) I suggest to justify or explain some settings of the experiments.
Such as: n=30 (line 564).
Are 60 samples sufficient for an accurate and reliable statistical analysis?

(4) More details about the results are needed.
From Table2 to Table 5, it is observed that the same MR with different statistical tests yield varying testing effectiveness. It seems that t-test are more effective than Chi-square Test and maximum voting. Are there any indications for deciding which statistical test should be used?

(5) Others.
Some images are blur, for example, figure 1 and figure 6
Line 783, “testing” is duplicate.

---

## Round 0.2 · Minor Revisions

· Academic Editor

Minor Revisions

Dear Authors,

Please clarify the the statement identified by Reviewer #3

M.P.

Reviewer 3 ·

Basic reporting

None

Experimental design

None

Validity of the findings

None

Additional comments

(1) "If the output for the source test case input differs from the output for the follow-up test case input, the MR::is said:: to:: be:::::::::: dissatisfied"
This statement is not a general description about MR dissatification. We check the source and follow-up outputs againts the MR: if they violate the specified relationships, then the MR is dissatisfied.
Equality is only of the MR relationship.
Maybe in this study, MRs employ equility output relationships, but this is only one of the cases.
I suggest the author to clarify the retionale of MT clearly.

---

## Round 0.3 · accepted · Accept

· Academic Editor

Accept

Dear Authors,

addressed all of the reviewers' comments and the manuscript is ready for publication.

M.P.